# Safety verification for polysorbate 20, pharmaceutical excipient for intramuscular administration, in Sprague-Dawley rats and New Zealand White rabbits

Junhyung Kim[1‡], Seongsung Kwak[1,2‡], Mi-Sun Park[1], Chang-Hoon Rhee[3], Gi-Hyeok Yang[1], Jangmi Lee[2], Woo-Chan Son[4‡*], Won-ho Kang[1‡*]

1 Gwanggyo R&D Center, Medytox Inc., Suwon-si, Gyeonggi-do, Korea, 2 Department of Medical Science, Asan Medical Institute of Convergence Science and Technology, Asan Medical Center, University of Ulsan College of Medicine, Songpa-gu, Seoul, Korea, 3 Osong R&D Center, Medytox Inc., Cheongju-si, Chungcheongbuk-do, Korea, 4 Department of Pathology, Asan Medical Center, Songpa-gu, Seoul, Korea

‡ JK and SK contributed equally to this work and served as co-first authors. WhK and WCS also contributed equally to this work and served as corresponding authors.
* whkang@medytox.com (WhK); wcson32@hanmail.net (WCS)

**Data Availability Statement:** All relevant data are within the manuscript and its Supporting information files.

## Abstract

Human serum albumin (HSA) has been widely used as a pharmaceutical excipient in Botulinum toxin serotype A (BoNT/A) products that are indicated for use in therapeutics and cosmetics. However, HSA as a human-derived material has some concerns, such as the potential risk of transmission of infectious agents, an insufficient supply, and difficulty in maintaining a certain quality. For those reasons, newly developed BoNT/A products (CORETOX®, Medytox, Inc., Republic of Korea) contained polysorbate 20, a non-human-derived excipient, to replace the HSA. However, most safety studies of polysorbate 20 have been conducted with non-invasive routes of administration, and thus there are a few studies on the safety of polysorbate 20 when administered intramuscularly. To secure the *in vivo* safety profile of polysorbate 20, a four-week repeated intramuscular dose toxicity study (0.02, 0.1, and 0.4 mg/kg, one injection every two weeks for a total of three injections) was conducted in 66 Sprague-Dawley (SD) rats. An intradermal irritation study was further conducted with 18 New Zealand White (NZW) rabbits. The toxicological evaluation of HSA (0.06 and 0.12 mg/kg) was also carried out as a comparative substance. Systemic and local toxicities were not observed in any of the SD rats or NZW rabbits based on clinical signs, body weight, hematology, clinical biochemistry, macroscopic findings on necropsy, histopathology of the injection site, and allergic reactions. The current study suggested that intramuscular administration of polysorbate 20 was considered to be safe at a level similar to that of HSA, which has an *in vivo* safety profile accumulated over the years. This provided the basis for the *in vivo* safety profile of polysorbate 20 administered intramuscularly and the scientific reliability of the use of polysorbate 20 as an alternative to HSA, which is used as an excipient for various pharmaceuticals in terms of its safety.

**Funding:** This study was supported by the Technology Innovation Program, establishment of risk management platform with aim of reduce attrition of new drugs and its service [grant number: 10067737, 2016], funded by the Ministry of Trade, Industry & Energy in Republic of Korea (WCS). The funders had no role in study design, data collection and analysis, decision to publish, or preparation of the manuscript. The authors [JK, SK, MSP, CHR, GHY, WhK] are employed by a commercial company, Medytox Inc., which only provided support in the form of salaries for authors, but did not have any additional role in the study design, data collection and analysis, decision to publish, or preparation of the manuscript. The specific roles of these authors are articulated in the 'author contributions' section.

**Competing interests:** The authors have declared that no competing interests exist. Furthermore, the authors [JK, SK, MSP, CHR, GHY, WhK] employed by a commercial company, Medytox Inc., confirm that this commercial affiliation does not alter their adherence to PLOS ONE policies on sharing data and materials.

# Introduction

Human serum albumin (HSA), the most abundant protein in human plasma, has the functions of regulating the colloid osmotic pressure, binding to and transporting various molecules, antioxidant action, anticoagulant action, regulating membrane permeability, and a direct neuroprotective action [1, 2]. Therefore, HSA has been used for therapeutic purposes in patients with hemorrhage, hypovolemia, and hypoalbuminemia since about the 19[th] century [2]. In addition to therapeutic purposes, it is used as an excipient for various pharmaceuticals in biomedical aspects [1, 3].

Most botulinum toxin serotype A (BoNT/A) products that are indicated for use in therapeutics and cosmetics also use HSA as an excipient [4, 5]. In BoNT/A products, HSA plays a role in stabilizing the neurotoxic proteins during manufacturing, transportation, and administration and in preventing the aggregation of neurotoxic proteins [6]. However, there are several concerns with HSA, although it is produced under strict regulations and has a safety profiling accumulated over the years [2, 6]. Since the production of HSA mainly relies on human blood donation, the quality of HSA could differ depending on the population of the human blood donors (race or geographical distribution) [7]. Furthermore, the market demand for HSA has dramatically increased (over 500 tons), and it is one of the most widely used biopharmaceutical solutions today [7, 8]. This could lead to a worldwide shortage of HSA in the market [2]. Finally, the use of HSA has a theoretical risk of spreading blood-derived pathogens [6, 9]. Currently, the transmission of viral diseases, such as hepatitis B virus, human immunodeficiency virus, and West Nile virus are well controlled through strict management of HSA, but there are still concerns about the transmission of prion-derived variant Creutzfeldt-Jakob disease [10, 11].

For those reasons, Medytox, Inc. (Republic of Korea) has recently developed a complexing protein-free BoNT/A product (CORETOX®) that uses polysorbate 20, a non-human-derived excipient, to replace the HSA [12, 13]. Polysorbates, a series of polyoxyethylenated sorbitan esters, are hydrophilic and nonionic surfactants that have been widely used as a food additive and stabilizer in pharmaceuticals [14, 15]. Among them, polysorbate 20, called polyoxyethylene-20-sorbitan monolaurate, has 20 repeating units of polyethylene glycol. It is mainly used as a direct/indirect additive in food and emulsifiers/surfactants in cosmetics that come into direct contact with the human skin, such as shampoo, bath soap, skin cleansing products, and hair spray [15]. It has also been used as an excipient in pharmaceuticals as a stabilizer, particularly in tablet and ophthalmic solutions. Hence, most of the safety studies on polysorbate 20 have been conducted on oral toxicity and skin irritation [14]. As a result, it was found that excessive dose of polysorbate20 could induce mild/temporary eye irritation and skin irritation [14–17]. In addition, when administered orally, it adversely affects the gastro-intestinal tract and, in severe cases, could cause death [14, 16]. However, there are a few studies on the safety of polysorbate 20 administered by other routes, especially intramuscular administration, which is the route of BoNT/A injection.

The objective of this study was to verify the *in vivo* safety properties of polysorbate 20 when administered intramuscularly. First, a four-week repeated intramuscular dose toxicity study in Sprague-Dawley (SD) rats (one injection every two weeks for a total of three injections) was carried out to evaluate its systemic toxicity. Second, an intradermal irritation study in New Zealand White (NZW) rabbits was carried out to evaluate local toxicity and allergic reactions. It was expected the current study could enable the evaluation of the safety of polysorbate 20 for intramuscular administration and would provide the basis for the *in vivo* safety profile of polysorbate 20 as a pharmaceutical excipient.

## Materials and methods

### Animals and facility

This study was reviewed and approved by the Institutional Animal Care and Use Committee of Medytox, Inc. (A-2019-009) and the Korea Testing & Research Institute (KTR, IAC2019-1500). In addition, the study design was established based on the ARRIVE guidelines [18].

In the four-week repeated intramuscular dose toxicity study performed at Medytox R&D Center (Republic of Korea), a total of 66 SD rats, consisting of 30 females (130.7 ~ 146.2g) and 36 males (163.5 ~ 187.2g), were used (Table 1). Healthy five-week-old SD rats were purchased from Orient Bio, Inc. (Republic of Korea). Upon receipt, all animals were macroscopically examined and acclimated for 7 days to the laboratory conditions (a specific pathogen-free animal room; temperature, 23 ± 3˚C; humidity, 55 ± 15%; ventilation, 15 air changes per hour; light/dark cycle, 12 hours; light intensity, 150 ~ 300 Lux; polycarbonate cage; 2 ~ 3 animals per cage). Animals were divided into six groups with 11 animals (six male and five female SD rats) in each group. Group 1 (control group) were administered 0.9% saline every two weeks. Group 2, 3, and 4 were administered 0.02, 0.1, and 0.4 mg/kg of polysorbate 20 every two weeks, respectively. Group 5 and 6 were administered 0.06 and 0.12 mg/kg of HSA every two weeks, respectively.

In the intradermal irritation study performed at KTR (Republic of Korea), a total of 18 male NZW rabbits (2307.7 ~ 2947.8 g) were used (Table 1). Healthy three-month-old rabbits

**Table 1. Study design of four-week repeated intramuscular dose toxicity study in Sprague-Dawley rats and intradermal irritation study in New Zealand white rabbits.**

| Study | Species | Group | Substance | Dose volume | Dose | Route and Frequency of administration | Number of animals (male/female) |
|---|---|---|---|---|---|---|---|
| **Four-week repeated intramuscular dose toxicity study** | **Sprague-Dawley Rats** | G1 | Control (0.9% saline) | 0.2 mL/kg | 0 mg/kg | Intramuscular injection (Gastrocnemius muscle) Once per 2 weeks (Total 3 times) | 11 (6/5) |
| | | G2 | Polysorbate 20 | | 0.02 mg/kg | | 11 (6/5) |
| | | G3 | Polysorbate 20 | | 0.1 mg/kg | | 11 (6/5) |
| | | G4 | Polysorbate 20 | | 0.4 mg/kg | | 11 (6/5) |
| | | G5 | Human serum albumin | | 0.06 mg/kg | | 11 (6/5) |
| | | G6 | Human serum albumin | | 0.12 mg/kg | | 11 (6/5) |
| | | **Total** | | | | | **66 (36/30)** |
| **Intradermal irritation study** | **New Zealand White rabbits** | G1 | Control (0.9% saline) | 0.4 mL/site (Total 5 sites) | 0 mg/kg/site | Intradermal injection (Dorsal back skin) Once (Total 1 time) | 3 (3/0) |
| | | G2 | Polysorbate 20 | | 0.02 mg/kg/site | | 3 (3/0) |
| | | G3 | Polysorbate 20 | | 0.1 mg/kg/site | | 3 (3/0) |
| | | G4 | Polysorbate 20 | | 0.4 mg/kg/site | | 3 (3/0) |
| | | G5 | Human serum albumin | | 0.06 mg/kg/site | | 3 (3/0) |
| | | G6 | Human serum albumin | | 0.12 mg/kg/site | | 3 (3/0) |
| | | **Total** | | | | | **18 (18/0)** |

were purchased from DooYeol Biotech (Republic of Korea). Upon receipt, all animals were macroscopically examined and acclimated for 6 days to the laboratory conditions (specific pathogen-free animal room; temperature, 20.3 ~ 21.9°C; humidity, 54.5 ~ 64.9%; ventilation, 10~ 20 air changes per hour; light/dark cycle, 12 hours; light intensity, 150 ~ 300 Lux; stainless steel cage; 1 animal per cage). Animals were divided into six groups with 3 male NZW rabbits in each group. Group 1 (control group) were administered 0.9% saline once. Group 2, 3, and 4 were administered 0.02, 0.1, and 0.4 mg/kg/site of polysorbate 20 once, respectively. Group 5 and 6 were administered 0.06 and 0.12 mg/kg/site of HSA once, respectively.

## Vehicle, positive control substance, and test substance

Saline (0.9% NaCl, DAI HAN PHARM. Co., Republic of Korea) was used as the vehicle in this study conducted in SD rats and NZW rabbits. HSA (Green Cross Corp., Republic of Korea) was selected as the comparative substance for the polysorbate 20 (Merck, Germany) which was the test substance of the current study. In the four-week repeated intramuscular dose toxicity study, the dose of HSA was calculated by multiplying the animal equivalent dose calculation factor (6.2) and the content of HSA in most BoNT/A products, 0.01 and 0.02 mg/kg (0.5 and 1.0 mg/vial, 50 kg human) [4, 19]. As a result, 0.06 and 0.12 mg/kg were selected as the dose of HSA. The dose of polysorbate 20 was selected based on the content of polysorbate 20 in COR-ETOX® (1.0 mg/vial, 60 kg human) and the human dosage of CORETOX® (0.2 ~ 3.6 vial) [12, 13]. The dose of polysorbate 20 was also calculated in the same manner and 0.02, 0.1, and 0.4 mg/kg were selected as the dose of polysorbate 20.

Doses of polysorbate 20 were calculated in the same way, and 0.02, 0.1 and 0.4 mg/kg were selected.

The same dose of HSA and polysorbate 20 used in the four-week repeated intramuscular dose toxicity study was applied in the intradermal irritation study in NZW rabbits. Detailed information is provided in Table 1.

## *In vivo* administration

In the four-week repeated intramuscular dose toxicity study in SD rats, all substances (saline, HSA, polysorbate 20) were administered intramuscularly. The dosing formulation was prepared on the day of administration using saline (HSA: 0.3, 0.6 mg/mL; polysorbate 20: 0.1, 0.5, 2.0 mg/mL). The dose volume was selected as 0.2 mL/kg and the individual dose volume for each animal was calculated based on the body weight of the animal on the day of administration [20]. All substances were injected once every two weeks for four weeks (a total of 3 times; days 1, 15, 29) into one site of the gastrocnemius muscle of the right hindlimb. The dosing schedule (interval and frequency) was set based on the clinical application schedule (total 1 time, at least three month dosing interval) and the administration site (gastrocnemius muscle) was selected based on the site generally used to evaluate the effectiveness of BoNT/A in rodents [12, 21–23].

In the intradermal irritation study in NZW rabbits, all substances were administered via the intradermal route. The dosing formulation was prepared on the day of administration using saline (HSA: 0.15, 0.3 mg/mL; polysorbate 20: 0.05, 0.25, 1.0 mg/mL). The dose volume was selected as 0.4 mL/kg/site (total 2 mL/kg/head) and the individual dose volume for each animal was calculated based on the body weight of animal on the day of administration [20]. Before administration, the back skin of the rabbit was depilated of hair. All substances were then injected once into five sites (three left side and two right side) on the back skin (total 1 time; day 1) based on the guideline of the International Organization for Standardization (ISO) [24].

## Clinical observation and body weights measurement

In the four-week repeated intramuscular dose toxicity study in SD rats, all animals were observed for clinical signs and general condition once daily throughout the study period (days 1 ~ 29). Their body weights were measured once a week (total five times), including the day of administration and at autopsy.

In the intradermal irritation study in NZW rabbits, clinical observation was conducted once daily throughout the study period (days 1 ~ 29). Furthermore, skin reactions at the administration site, including the degree of erythema, crust formation, and edema formation, were observed once daily based on the scoring system for skin reactions of the ISO [24]. Body weights were measured once a week (total five times), including on the day of administration and at autopsy.

## Hematology, clinical biochemistry, and allergic reaction test

All animals (SD rats and NZW rabbits) were fasted overnight before the last day of the study. The SD rats were anesthetized with isoflurane and blood samples were collected from their abdominal vein. In the NZW rabbits, blood samples were collected from the marginal ear vein without anesthesia.

Approximately 1 mL of the blood sample of all animals was put into a tube containing EDTA. After that, the following parameters were analyzed in KTR using a hematological auto-analyzer (ADVIA 120, SIEMENS, Germany): white blood cell count (WBC), differential leucocyte count (neutrophils; lymphocytes; monocytes; eosinophils; basophils), red blood cell count (RBC), hemoglobin (HGB), hematocrit (HCT), RBC indices (mean corpuscular volume, MCV; mean corpuscular hemoglobin, MCH; mean corpuscular hemoglobin concentration, MCHC), reticulocyte, and platelets.

The blood samples (except those used in the hematological analysis) were centrifuged at 3,000 rpm for 10 minutes to obtain the serum. The following parameters were then examined in KTR using an automatic analyzer (TBA-120FR, TOSHIBA, Japan): total protein, albumin, albumin/globulin ratio, total bilirubin, total bile acid, alkaline phosphatase (ALP), aspartate aminotransferase (AST), aspartate aminotransferase (AST), alanine aminotransferase (ALT), gamma glutamyltransferase (GGT), creatinine, blood urea nitrogen (BUN), total cholesterol, triglycerides (TG), glucose, calcium, inorganic phosphorus, creatine phosphokinase, cholinesterase, sodium, potassium, and chloride. Among these, total bile acid and cholinesterase were only analyzed for the NZW rabbits.

In addition, the levels of serum IgE and histamine of all animals were measured using an enzyme-linked immunosorbent assay (ELISA) kit.

## Necropsy and histopathology

After blood sampling from the SD rats and NZW rabbits, autopsies were performed on all animals at day 29 according to the AVMA Guidelines for the Euthanasia of Animals [25]. All SD rats were euthanized by exsanguination from the posterior vena cava and abdominal aorta under isoflurane anesthesia. All NZW rabbits were euthanized by intravenous injection of T-61 solution (MSD animal health, Republic of Korea). At necropsy, their external appearance, internal organs, and injection sites were observed macroscopically. The injection site (one site of the gastrocnemius muscle of the right hindlimb for the SD rats and the five sites of the back skin for the NZW rabbits) were collected and stored in 10% neutral buffered formalin. Slides of all preserved tissues for histopathology were prepared through general tissue processing, including dehydration and paraffin embedding. All slides were then stained with hematoxylin and eosin and examined microscopically.

## Statistical analysis

The Kruskal-Wallis test was performed to compare toxicological indicators, including body weights, hematology level, clinical biochemistry level, and serum IgE and histamine levels among the groups of SD rats and NZW rabbits. In the case of body weight, since there was a difference in the mean body weight among the groups before the first dose administration, the statistical analysis was performed using the ratio of body weight at the time of the measurement to the body weight before the first administration. The significance of intergroup difference between the control and administered groups was assessed using Dunn's Rank Sum test. All statistical analyses were performed using GraphPad Prism 8.0 (GraphPad Software, Inc., San Diego, CA) and a *p*-value of < 0.05 was accepted to indicate statistical significance. Data are represented as mean ± standard error of the mean (SEM).

## Results

### Mortality and clinical signs

In the four-week repeated intramuscular dose toxicity study, all SD rats remained well throughout the study and showed no clinical signs of disease, pain or distress. In the intradermal irritation study, no deaths or abnormal findings were observed in all NZW rabbits. Skin reactions, including erythema, eschar formation, and edema formation, were not observed regardless of the administered substance.

### Body weights

For both the SD rats and NZW rabbits, their body weight was measured once a week for a total of five times. Individual body weights of all animals by time point are presented in the S1 and S2 Tables.

In the four-week repeated intramuscular dose toxicity study, a gradual body weight gain of the SD rats was observed regardless of the administered substance (Fig 1a). Compared to the body weight before the first administration, all male SD rats had a body weight of 138.4% at day 8, 173.6% at day 15, 200.4% at day 22, and 225.0% at day 29, while all female SD rats had a body weight of 132.2% at day 8, 153.1% at day 15, 173.1% at day 22, and 187.3% at day 29. There was no significant increase or decrease of the body weight of the SD rats administered polysorbate 20 and HSA compared to the control group at any of the time points.

In the intradermal irritation study, a gradual body weight gain of the NZW rabbits was also observed regardless of the administered substance (Fig 1b). Compared to their body weight before administration, all NZW rabbits had a weight of 105.0% at day 8, 111.0% at day 15, 116.7% at day 22, and 120.8% at day 29. There was also no significant difference in body weight of the NZW rabbits administered polysorbate 20 and HSA compared to the control group at any of the time points.

### Hematology, clinical biochemistry, and serum IgE and histamine levels

Hematological tests and clinical biochemistry tests were performed on all SD rats and NZW rabbits. Individual results are presented in the S3 and S4 Tables. In both male and female SD rats, there were no polysorbate 20 or HSA related changes in the hematology parameters. In clinical biochemistry, compared to the control group, the albumin/globulin ratio significantly decreased in male group 6 ($p < 0.05$), the level of glucose significantly decreased in male group 3, 5, and 6 ($p < 0.05$, respectively), and the level of sodium significantly increased in male group 4 ($p < 0.01$) and 6 ($p < 0.05$) (Table 2). In females, the level of total bilirubin significantly decreased in group 2 and 4 ($p < 0.05$, respectively) and the level of chloride significantly

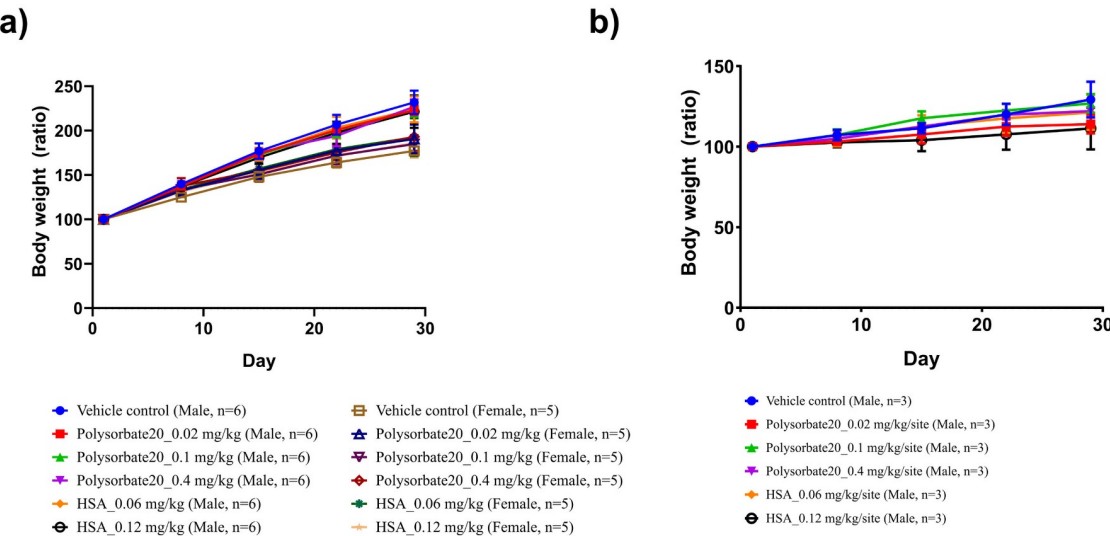

**Fig 1. Body weight changes in a four-week repeated intramuscular dose toxicity study in Sprague-Dawley (SD) rats and an intradermal irritation study in New Zealand White (NZW) rabbits.** Body weight changes measured once a week in a) SD rats and b) NZW rabbits, for a total of five times during the study period. In SD rats, all substances were injected once every two weeks for four weeks (total three times; days 1, 15, 29) into one site of the gastrocnemius muscle of the right hindlimb. Neither male nor female rat groups showed significant differences in body weight on days 8, 15, 22, and 29 (Kruskal-Wallis test). In NZW rabbits, all substances were injected once into five sites of the back skin (total one time; day 1). There was no significant difference in body weight on days 8, 15, 22, and 29 in all groups (Kruskal-Wallis test).

increased in group 2 ($p < 0.05$) and 3 ($p < 0.01$), compared to the control group (Table 3). Furthermore, in the allergic reaction test (ELISA analysis of serum IgE and histamine), significantly higher serum histamine levels were found in the female group 5 and 6 compared to the control group ($p < 0.05$).

In NZW rabbits, there were no changes in the hematology and clinical biochemistry parameters in the polysorbate 20 or HSA administered animals. There was also no significant change in the level of serum IgE and histamine in NZW rabbits administered polysorbate 20 and HSA compared to the control group.

## Necropsy and histopathology

During the necropsy on all SD rats, no polysorbate 20 or HSA related macroscopic findings were observed such as changes in external appearance, internal organs, and administration sites. In histopathological examination of the administration site, mild macrophage infiltration and neovascularization were observed with a similar degree and frequency in all administered groups, including the control group. In addition, inflammatory cells infiltration, fat infiltration, giant cells, and necrosis were observed only in some individuals regardless of the administered substance and dose (Fig 2).

In the case of the NZW rabbits, macroscopic findings at necropsy and histopathological findings at the sites of administration were not observed regardless of the administered substance (Fig 3).

## Discussion

Polysorbate 20 was approved by the regulatory authorities as an additive in foods, cosmetics, eye drops, and tablets in various countries [14, 15]. Therefore, most clinical and nonclinical safety studies of polysorbate 20 have been conducted using non-invasive routes of

**Table 2. Hematology, clinical biochemistry, and serum IgE and histamine levels in male Sprague-Dawley (SD) rats.**

| Category | | Male | | | | | |
|---|---|---|---|---|---|---|---|
| | | G1 | G2 | G3 | G4 | G5 | G6 |
| Hematological findings | RBC ($10^6$cells/μL) | 7.37±0.13 | 7.38±0.14 | 7.31±0.11 | 7.48±0.14 | 7.50±0.10 | 7.37±0.14 |
| | HGB (g/dL) | 14.52±0.20 | 14.57±0.22 | 14.57±0.31 | 14.58±0.23 | 14.7±0.12 | 14.37±0.24 |
| | HCT (%) | 46.30±0.40 | 46.15±0.68 | 45.87±0.85 | 46.22±0.57 | 46.42±0.36 | 45.67±0.89 |
| | MCV (fL) | 62.95±0.81 | 62.57±0.55 | 62.73±0.56 | 61.82±0.57 | 61.95±1.00 | 62.02±0.09 |
| | MCH (pg) | 19.72±0.27 | 19.73±0.16 | 19.90±0.17 | 19.50±0.31 | 19.57±0.27 | 19.50±0.13 |
| | MCHC (g/dL) | 31.33±0.19 | 31.60±0.07 | 31.72±0.24 | 31.55±0.23 | 31.62±0.12 | 31.48±0.19 |
| | RETIC ($10^9$cells/μL) | 218.7±8.4 | 213.2±9.4 | 225.7±11.4 | 238.7±4.2 | 198.7±11.3 | 222.8±10.0 |
| | PLT ($10^3$ cells/μL) | 1156±63 | 1168±55 | 1161±66 | 1210±39 | 1228±28 | 1170±47 |
| | WBC ($10^3$ cells/μL) | 10.81±1.14 | 10.90±0.81 | 9.68±0.95 | 8.56±0.44 | 7.15±0.76 | 8.14±0.87 |
| Clinical biochemical findings | TP (g/dL) | 5.90±0.06 | 6.07±0.08 | 5.97±0.11 | 6.00±0.09 | 6.10±0.09 | 6.13±0.10 |
| | ALB (g/dL) | 3.85±0.04 | 3.88±0.03 | 3.83±0.06 | 3.85±0.03 | 3.97±0.04 | 3.88±0.06 |
| | A/G ratio | 1.88±0.04 | 1.78±0.03 | 1.78±0.04 | 1.80±0.04 | 1.87±0.04 | 1.73±0.02* |
| | T-BIL (mg/dL) | 0.04±0.00 | 0.05±0.00 | 0.04±0.00 | 0.03±0.00 | 0.03±0.00 | 0.05±0.00 |
| | ALP (U/L) | 596.3±33.6 | 711.0±57.2 | 532.3±38.5 | 577.2±30.6 | 560.2±35.6 | 647.5±47.3 |
| | AST (U/L) | 90.33±6.35 | 119.2±30.2 | 92.67±4.34 | 81.00±4.46 | 106.5±4.8 | 101.5±9.7 |
| | ALT (U/L) | 28.33±3.46 | 45.50±16.37 | 27.17±2.85 | 22.83±1.56 | 29.00±2.66 | 26.17±2.44 |
| | CREA (mg/dL) | 0.44±0.01 | 0.45±0.02 | 0.46±0.02 | 0.44±0.02 | 0.40±0.01 | 0.42±0.01 |
| | BUN (mg/dL) | 11.78±0.60 | 12.60±1.00 | 11.93±0.54 | 10.42±0.74 | 11.17±0.48 | 10.90±0.41 |
| | T-CHO (mg/dL) | 71.00±3.39 | 78.50±4.65 | 76.33±4.26 | 72.33±5.33 | 65.83±5.68 | 82.00±3.01 |
| | TG (mg/dL) | 80.33±7.06 | 99.67±13.84 | 112.5±15.3 | 96.83±13.26 | 81.33±9.54 | 93.67±17.89 |
| | GLU (mg/dL) | 127.7±2.9 | 111.0±4.2 | 103.5±3.7* | 117.0±7.9 | 99.33±3.33* | 99.50±6.82* |
| | CA (mg/dL) | 10.15±0.11 | 10.23±0.05 | 10.13±0.10 | 10.32±0.06 | 10.00±0.10 | 10.23±0.14 |
| | IP (mg/dL) | 8.90±0.18 | 8.68±0.17 | 8.63±0.15 | 8.78±0.13 | 9.02±0.09 | 8.90±0.11 |
| | GGT (IU/L) | 0.39±0.16 | 0.34±0.18 | 0.69±0.26 | 0.34±0.16 | 0.19±0.09 | 0.48±0.26 |
| | CK (U/L) | 529.7±45.1 | 529.7±29.3 | 578.5±60.3 | 452.5±39.0 | 745.5±50.4 | 652.3±79.4 |
| | Na (mmol/L) | 143.0±0.3 | 144.4±0.5 | 143.6±0.3 | 145.2±0.4** | 144.3±0.4 | 144.9±0.4* |
| | K (mmol/L) | 4.68±0.08 | 4.68±0.06 | 4.76±0.08 | 4.87±0.12 | 4.95±0.05 | 4.89±0.10 |
| | Cl (mmol/L) | 101.4±0.3 | 102.8±0.5 | 101.4±0.8 | 103.1±0.5 | 101.5±0.6 | 102.3±0.6 |
| ELISA analysis | IgE (ng/mL) | 0.233±0.014 | 0.270±0.001 | 0.246±0.015 | 0.181±0.008 | 0.200±0.016 | 0.188±0.004 |
| | Histamine (ng/mL) | 0.480±0.040 | 0.383±0.038 | 0.402±0.034 | 0.423±0.026 | 0.519±0.052 | 0.390±0.040 |

Data are presented as mean ± SEM.

All data were analyzed using Kruskal-Wallis test and the significance of intergroup difference between the control and administered groups was assessed using Dunn's Rank Sum test.

*vs. Control group: $p < 0.05$,

**vs. Control group: $p < 0.01$

administration (e.g., oral, skin, and ophthalmic administration) [16, 17]. In particular, as a food additive, polysorbate 20 has various *in vivo* safety data, including *in vitro* genotoxicity and DNA reactivity based on structural alerts [26]. Recently, polysorbate 20, which was one of the excipients of CORETOX® approved by the Ministry of Food and Drug Safety (MFDS, Republic of Korea), has been used in humans via an intramuscular route of administration [12]. However, there are few studies on the safety of polysorbate 20 when administered intramuscularly. In a study conducted in four monkeys, it was reported that intramuscular injection of 275 mg/kg of polysorbate20 for 20 days did not cause toxicity in the liver and kidney [27]. This study did not evaluate clinical signs, body weight, or hematology and clinical

**Table 3. Hematology, clinical biochemistry, and serum IgE and histamine levels in female Sprague-Dawley (SD) rats.**

| Category | | Female | | | | | |
|---|---|---|---|---|---|---|---|
| | | G1 | G2 | G3 | G4 | G5 | G6 |
| Hematological findings | RBC ($10^6$cells/µL) | 7.14±0.20 | 6.72±0.17 | 7.06±0.08 | 7.30±0.09 | 7.47±0.22 | 7.30±0.13 |
| | HGB (g/dL) | 14.08±0.30 | 13.78±0.31 | 13.96±0.20 | 14.58±0.12 | 14.74±0.28 | 14.52±0.28 |
| | HCT (%) | 42.54±0.88 | 42.08±1.01 | 42.06±0.46 | 44.42±0.16 | 44.60±0.88 | 44.40±0.76 |
| | MCV (fL) | 59.70±0.89 | 62.70±1.13 | 59.60±0.91 | 60.92±0.69 | 59.84±0.92 | 60.84±0.60 |
| | MCH (pg) | 19.76±0.22 | 20.46±0.06 | 19.78±0.36 | 20.00±0.26 | 19.76±0.29 | 19.88±0.19 |
| | MCHC (g/dL) | 33.12±0.27 | 32.70±0.50 | 33.20±0.15 | 32.84±0.17 | 33.00±0.28 | 32.70±0.15 |
| | RETIC ($10^9$cells/µL) | 156.9±10.2 | 170.9±16.1 | 151.9±14.7 | 169.0±5.2 | 143.1±13.7 | 159.8±7.6 |
| | PLT ($10^3$ cells/µL) | 995.6±167.9 | 858.2±201.0 | 1208±111 | 1164±78 | 1097±56 | 1217±63 |
| | WBC ($10^3$ cells/µL) | 3.48±0.43 | 4.24±0.97 | 5.62±1.00 | 5.88±1.13 | 4.71±0.35 | 4.38±0.37 |
| Clinical biochemical findings | TP (g/dL) | 6.10±0.05 | 6.06±0.14 | 6.10±0.23 | 6.04±0.04 | 6.02±0.09 | 6.20±0.13 |
| | ALB (g/dL) | 4.14±0.06 | 4.02±0.12 | 3.94±0.19 | 3.90±0.06 | 3.90±0.03 | 4.04±0.09 |
| | A/G ratio | 2.10±0.05 | 1.98±0.08 | 1.84±0.11 | 1.84±0.06 | 1.86±0.04 | 1.88±0.04 |
| | T-BIL (mg/dL) | 0.07±0.00 | 0.03±0.00* | 0.05±0.01 | 0.03±0.01* | 0.04±0.01 | 0.04±0.01 |
| | ALP (U/L) | 354.8±16.1 | 483.2±45.1 | 448.2±37.2 | 433.0±49.5 | 442.4±44.8 | 474.6±48.6 |
| | AST (U/L) | 83.00±2.51 | 96.60±6.17 | 92.60±13.1 | 92.00±2.00 | 82.00±4.32 | 84.20±7.04 |
| | ALT (U/L) | 26.20±0.80 | 27.00±2.55 | 31.20±5.00 | 25.40±2.38 | 33.20±2.06 | 29.60±1.96 |
| | CREA (mg/dL) | 0.43±0.02 | 0.46±0.03 | 0.43±0.01 | 0.41±0.02 | 0.39±0.01 | 0.43±0.02 |
| | BUN (mg/dL) | 19.88±2.57 | 15.58±1.01 | 15.94±0.88 | 13.46±1.22 | 14.62±0.86 | 16.86±1.08 |
| | T-CHO (mg/dL) | 86.80±9.28 | 85.80±3.81 | 87.00±4.55 | 86.60±6.59 | 86.40±3.82 | 97.80±9.41 |
| | TG (mg/dL) | 76.20±19.14 | 47.60±7.47 | 48.60±3.66 | 47.60±16.46 | 52.40±6.54 | 63.80±7.65 |
| | GLU (mg/dL) | 114.0±4.4 | 124.0±6.5 | 104.0±2.7 | 103.6±5.2 | 113.2±4.6 | 120.4±6.6 |
| | CA (mg/dL) | 10.04±0.10 | 9.96±0.13 | 10.16±0.20 | 10.08±0.14 | 10.28±0.04 | 10.44±0.10 |
| | IP (mg/dL) | 7.36±0.25 | 7.30±0.23 | 7.48±0.23 | 7.50±0.22 | 7.76±0.17 | 8.00±0.13 |
| | GGT (IU/L) | 0.34±0.09 | 0.17±0.07 | 0.21±0.13 | 0.32±0.14 | 0.20±0.13 | 0.43±0.12 |
| | CK (U/L) | 606.4±15.2 | 683.2±67.8 | 533.6±118.0 | 491.4±28.3 | 431.6±45.0 | 508.6±114.3 |
| | Na (mmol/L) | 140.8±0.3 | 141.6±0.5 | 141.7±0.7 | 141.9±0.3 | 141.1±0.3 | 141.2±0.3 |
| | K (mmol/L) | 4.44±0.07 | 4.29±0.14 | 4.50±0.16 | 4.44±0.09 | 4.54±0.12 | 4.43±0.22 |
| | Cl (mmol/L) | 100.2±0.3 | 102.5±0.4* | 103.0±0.6** | 101.6±0.9 | 102.3±0.4 | 101.5±0.3 |
| ELISA analysis | IgE (ng/mL) | 0.222±0.023 | 0.211±0.008 | 0.247±0.035 | 0.211±0.025 | 0.187±0.022 | 0.183±0.016 |
| | Histamine (ng/mL) | 0.253±0.030 | 0.311±0.043 | 0.385±0.027 | 0.257±0.034 | 0.570±0.128* | 0.510±0.063* |

Data are presented as mean ± SEM.

All data were analyzed using Kruskal-Wallis test and the significance of intergroup difference between the control and administered groups was assessed using Dunn's Rank Sum test.

*vs. Control group: $p < 0.05$,

** vs. Control group: $p < 0.01$

biochemistry, which are essential criteria for evaluating toxicity [28]. Therefore, the current study was conducted to verify the safety of intramuscular administration of polysorbate 20 through four-week repeated intramuscular dose toxicity in SD rats and an intradermal irritation study in NZW rabbits. This study is the first to fully evaluate the *in vivo* safety of polysorbate 20 when administered intramuscularly based on current guidelines of non-clinical safety research.

To evaluate the systemic toxicity of intramuscular injection of polysorbate 20, a four-week repeated dose toxicity study in SD rats was conducted. According to the MFDS, the clinical dosage of CORETOX®, which contains 1 mg of polysorbate 20 in one vial, ranged from 0.2

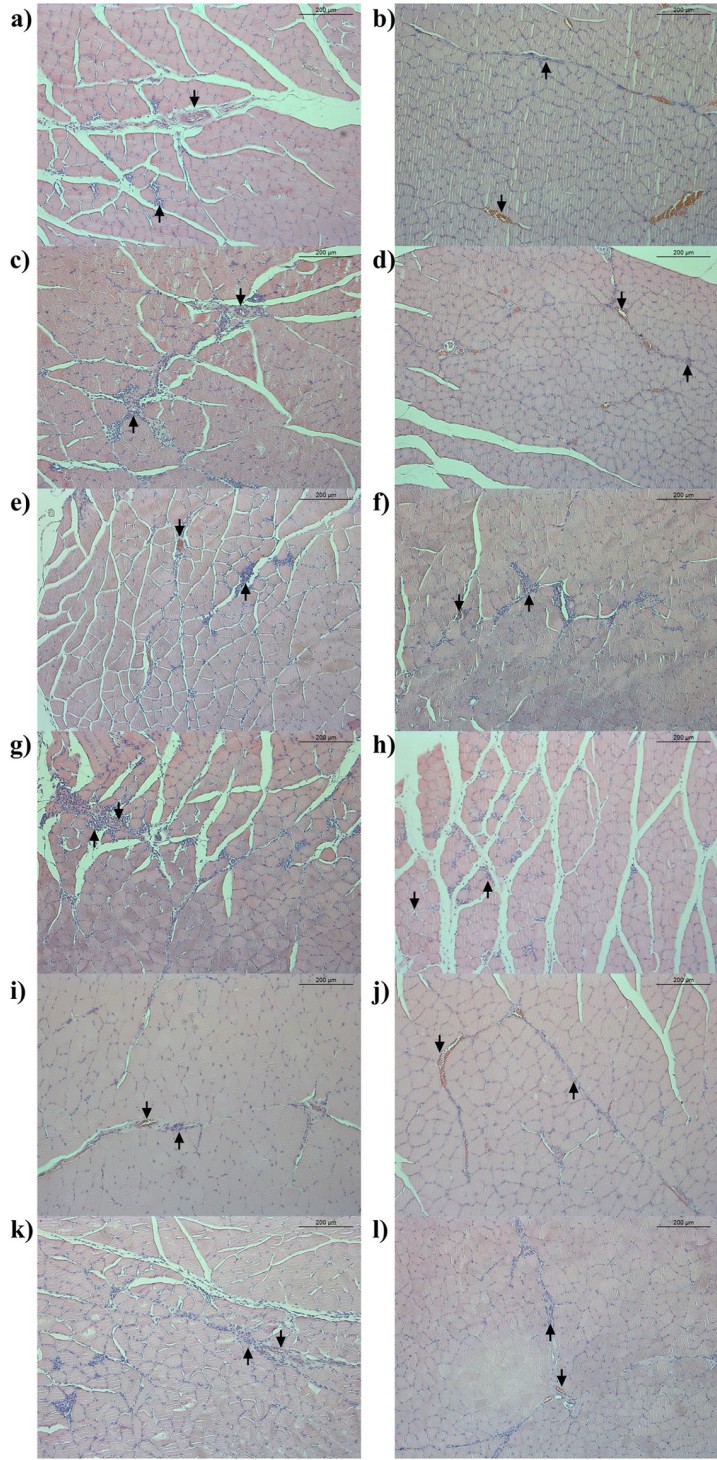

**Fig 2. Histopathological section of the administration site in the four-week repeated intramuscular dose toxicity study in Sprague-Dawley (SD) rats.** Hematoxylin and eosin stained histopathological section (original magnification 100x) of the gastrocnemius muscle of the SD rats: a) Control group, male; b) Control group, female; c) Polysorbate 20–0.02 mg/kg, male; d) Polysorbate 20–0.02 mg/kg, female; e) Polysorbate 20–0.1 mg/kg, male; f) Polysorbate 20–0.1 mg/kg, female; g) Polysorbate 20–0.4 mg/kg, male; h) Polysorbate 20–0.4 mg/kg, female; i) Human serum albumin-0.06 mg/kg, male; j) Human serum albumin-0.06 mg/kg, female; k) Human serum albumin-0.12 mg/kg, male; l) Human

serum albumin-0.12 mg/kg, female. Mild macrophage infiltration, neovascularization, inflammatory cells infiltration, and fat infiltration were observed with a similar degree and frequency in all administered groups. An arrow pointing up (↑) indicated macrophage infiltration and an arrow pointing down (↓) indicated neovascularization.

vial (treatment of forehead wrinkles) to 3.6 vials (stiff muscles in the upper limbs after a stroke) [12]. Thus, the dose (0.02, 0.1, and 0.4 mg/kg) of polysorbate 20 in the current study was established based on the therapeutic indication of CORETOX® in humans and the animal equivalent dose calculation factor [19]. In addition, 0.9% saline was set as the vehicle control substance, while 0.06 and 0.12 mg/kg doses of HSA, which has an *in vivo* safety profile

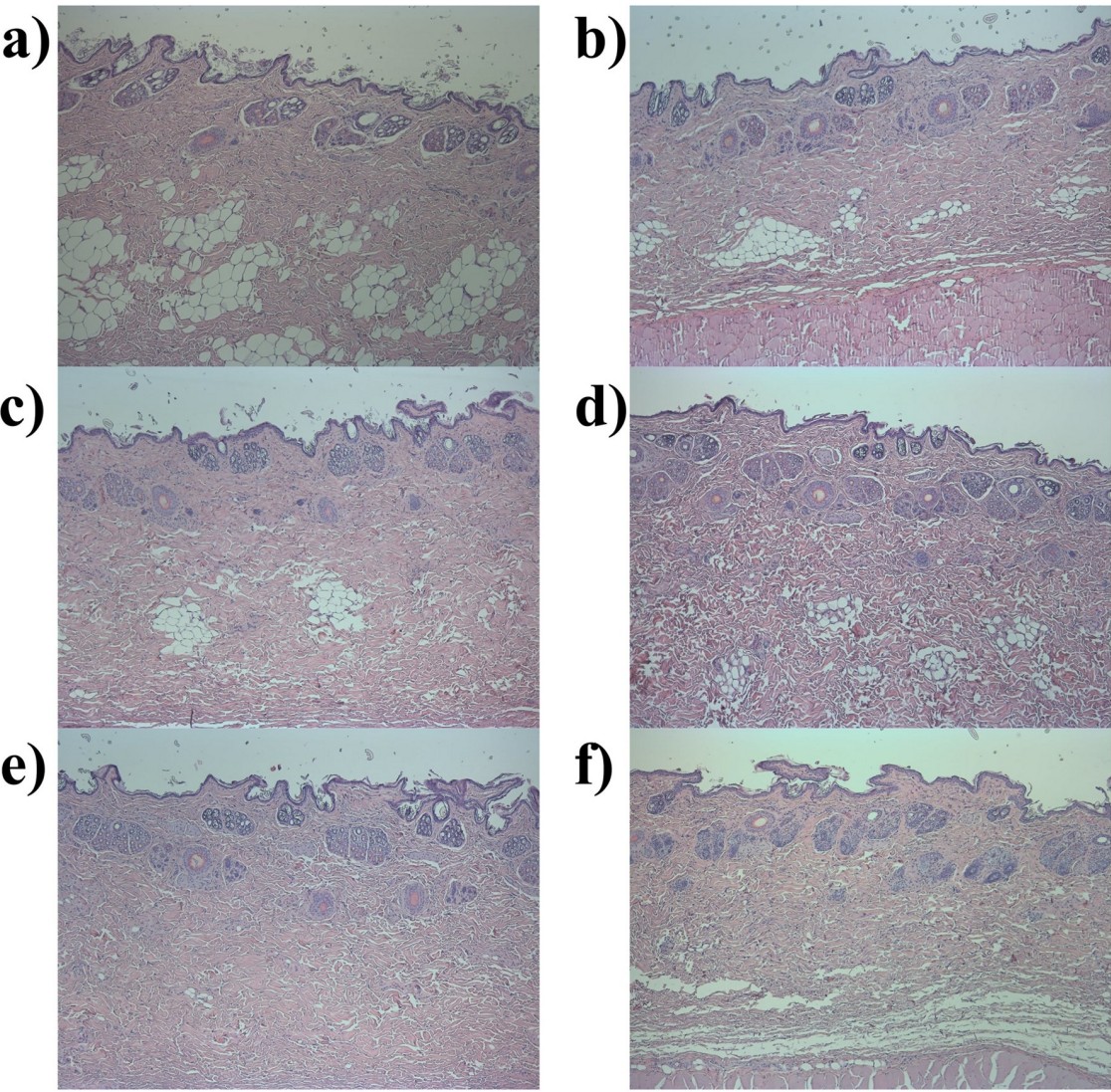

**Fig 3. Histopathological section of the administration site in the intradermal irritation study in New Zealand White (NZW) rabbits.** Hematoxylin and eosin stained histopathological section (original magnification 50x) of the dorsal back skin of the NZW rabbits: a) Control group; b) Polysorbate 20–0.02 mg/kg; c) Polysorbate 20–0.1 mg/kg; d) Polysorbate 20–0.4 mg/kg; e) Human serum albumin (HSA)-0.06 mg/kg; f) HSA-0.12 mg/kg, male. No histopathological findings were found at the site of administration in all administered groups.

accumulated over the years for intramuscular administration, was set as a comparative substance [2, 6].

Generally, mortality, body weight, and clinical signs have the most significance in repeated dose toxicity studies [29, 30]. Furthermore, changes in the levels of hematology and clinical biochemistry in laboratory animals could enable more accurate predictions of human toxicity [31, 32]. In the current study, adverse effects, including mortality, clinical signs, and macroscopic findings on necropsy, were not found at any dose level of HSA and polysorbate 20 administered during the study period in all SD rats. In addition, significant changes in the body weight and the level of hematology were not observed in the polysorbate 20 and HSA administered groups compared to the control group. In the clinical biochemistry test and allergy reaction test, significant decrease in albumin/globulin ratio ($p < 0.05$), glucose ($p < 0.05$), and total bilirubin ($p < 0.05$) and significant increase in sodium ($p < 0.05$ or $p < 0.01$), chloride ($p < 0.05$ or $p < 0.01$), and serum histamine ($p < 0.05$) were observed in the polysorbate 20 or HSA administered group compared to the control group. However, these changes had no toxicological significance as they were not dose-responsive and the levels were within the normal range for the same age of SD rats [33, 34]. This indicated that intramuscular administration of 0.02, 0.1, and 0.4 mg/kg dose of polysorbate 20 did not cause systemic toxicity in SD rats.

Histopathological examination of the administration site was conducted to observe the local toxicity of the polysorbate 20 in all SD rats. At the administration site, macrophage infiltration, neovascularization, giant cell, fibrosis, and necrosis were observed to a similar degree and frequency in all administered groups regardless of the administered substance. However, these findings were observed in most of the administered groups without a dose response, including the control group. Furthermore, inflammatory lesions could occur in the body of SD rats as a defense mechanism against intramuscular injection of any foreign substances [35]. Therefore, these findings were presumed to be due to the intramuscular injection rather than any effect of polysorbate 20. In other words, there were no toxicological changes at the site of administration after intramuscular administration of polysorbate 20 based on histopathology. Furthermore, serum IgE and histamine levels were measured to determine whether allergic reactions occurred after the administration of polysorbate 20 in all SD rats. In addition to clinical signs, increased levels of serum IgE and histamine are representative indicators of an allergic reaction [36–38]. In the current study, allergic reactions were not observed in response to the administration of polysorbate 20, which indicated that polysorbate 20 is not an allergen in SD rats.

However, despite its very low frequency, side effects in the allergy category have been reported in humans using shampoos (8.4% polysorbate 20, 2 cases among 5.88 million uses), cuticle softeners (2% polysorbate 20, 24 cases among 131 million use), and paste masks (2% polysorbate 20, 11 cases among 120.7 million uses) containing polysorbate 20 [14, 15]. Therefore, an additional intradermal irritation study was conducted in NZW rabbits to confirm that polysorbate 20 was not an allergen *in vivo*. Traditionally, the intradermal test is the gold standard method for diagnosing allergic reactions [39, 40]. If an immune response, such as erythema, crust formation, and edema formation, occurs at the administration site after intradermal administration of a substance, it could be concluded that the substance was an allergen to the administered subjects [24]. In the current study, immune responses of the administration site were not observed at all polysorbate 20 treated NZW rabbits after intradermal administration during the study period, which is the same as the results in the control group and the HSA administered group. Furthermore, in the polysorbate 20 administered group, toxicological changes based on the histopathological examination of the site of administration and increased levels of serum IgE and histamine were not observed. There were also no

toxicologically significant changes, such as changes in the body weight or the hematology and clinical biochemistry parameters in all rabbits administered polysorbate 20 and all toxicological values in NZW rabbits were within the normal range for the same age [41, 42]. These results of the intradermal irritation study indicated that polysorbate 20 is not an allergen in NZW rabbits, which was consistent with the results of the four-week repeated dose toxicity study in SD rats.

## Conclusions

The current study, a four-week repeated intramuscular dose toxicity study in SD rats and an intradermal irritation study in NZW rabbits, provides the basis for the *in vivo* safety profile of intramuscular administration of polysorbate 20, an excipient of newly developed BoNT/A products (CORETOX®). Polysorbate 20, which did not induce systemic and local toxicities in this study, was considered to be a safe substance for intramuscular administration, just like HSA for which a safety profile has been obtained over the years. In addition, since polysorbate 20 is a synthetic product rather than an animal-derived product, it has the advantages of a consistent quality of the product, ease of manufacture and supply, and a low risk of pathogen transmission compared to HSA. Therefore, the findings of this study provide the scientific reliability of the use of polysorbate 20 as an alternative to HSA, which is used as an excipient for most BoNT/A products, particularly in terms of its safety. Furthermore, this study suggest the possibility that polysorbate 20, which is mainly used as an additive in foods, cosmetics, eye drops, and tablets, could be used as a safe excipient for various pharmaceuticals administered intramuscularly.

## Supporting information

**S1 Table. Individual body weights of male (n: 36) and female (n: 30) Sprague-Dawley (SD) rats in the current study.**
(XLSX)

**S2 Table. Individual body weights of male New Zealand White (NZW) rabbits (n: 18) in the current study.**
(XLSX)

**S3 Table. Individual levels of hematology, clinical biochemistry, and serum IgE and histamine in Sprague-Dawley (SD) rats in the current study.**
(XLSX)

**S4 Table. Individual level of hematology, clinical biochemistry, and serum IgE and histamine in New Zealand White (NZW) rabbits in the current study.**
(XLSX)

## Acknowledgments

The authors thanks Do Yeon Lee and Min-Seo Choi, colleagues at the Medytox, Inc., for their helpful discussion and careful reading of the manuscript.

## Author Contributions

**Conceptualization:** Junhyung Kim, Seongsung Kwak, Woo-Chan Son, Won-ho Kang.

**Data curation:** Junhyung Kim, Seongsung Kwak, Mi-Sun Park, Chang-Hoon Rhee, Gi-Hyeok Yang, Jangmi Lee.

**Formal analysis:** Junhyung Kim, Seongsung Kwak.

**Funding acquisition:** Woo-Chan Son.

**Investigation:** Junhyung Kim, Seongsung Kwak, Mi-Sun Park, Chang-Hoon Rhee, Gi-Hyeok Yang.

**Methodology:** Junhyung Kim, Seongsung Kwak, Mi-Sun Park, Chang-Hoon Rhee, Gi-Hyeok Yang.

**Project administration:** Mi-Sun Park, Woo-Chan Son, Won-ho Kang.

**Resources:** Chang-Hoon Rhee, Gi-Hyeok Yang.

**Supervision:** Woo-Chan Son, Won-ho Kang.

**Validation:** Mi-Sun Park, Chang-Hoon Rhee, Gi-Hyeok Yang.

**Visualization:** Junhyung Kim.

**Writing – original draft:** Junhyung Kim, Seongsung Kwak.

**Writing – review & editing:** Junhyung Kim, Seongsung Kwak, Mi-Sun Park, Chang-Hoon Rhee, Gi-Hyeok Yang, Jangmi Lee, Woo-Chan Son, Won-ho Kang.

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
