## [Decision Letter · Decision Letter 0]

13 Apr 2021

PONE-D-21-03019

Safety verification for polysorbate 20, pharmaceutical excipient of newly developed Botulinum toxin serotype A products, in Sprague-Dawley rats and New Zealand White rabbits

PLOS ONE

Dear Dr. Kang,

Thank you for submitting your manuscript to PLOS ONE. After careful consideration, we feel that it has merit but does not fully meet PLOS ONE’s publication criteria as it currently stands. Therefore, we invite you to submit a revised version of the manuscript that addresses the points raised during the review process.

We look forward to receiving your revised manuscript.

Kind regards,

Arun K Yadav, Ph.D.

Academic Editor

PLOS ONE

Comments from Staff Editor:

Please address the following points as part of your revisions and include comments in your response to reviewers.

1) Why did you measure IgE levels and not for example HSA/PS specific IgG?

2) What was the rationale for using the gastrocnemius muscle for injection rather than the thigh muscles?

3) We recommend rephrasing Lines 202-203 to removed the toxicologists' bias of looking for death in a study ("all animals survived..") to "..all animals remained well throughout the study and showed no clinical signs of disease, pain or distress.".  Please also replace uses of the word 'symptoms' (what a patient reports) instead of 'signs' (what is observable). 

4) We note that the manuscript focuses mainly on the implications of this study for the development of cosmetics, and therefore request that you explicitly assert and emphasize the medical implications of this study in order to show the full range of applications of your findings.

Journal Requirements:

We note that one or more of the authors are employed by a commercial company: Medytox Inc.

2.1. Please provide an amended Funding Statement declaring this commercial affiliation, as well as a statement regarding the Role of Funders in your study. If the funding organization did not play a role in the study design, data collection and analysis, decision to publish, or preparation of the manuscript and only provided financial support in the form of authors' salaries and/or research materials, please review your statements relating to the author contributions, and ensure you have specifically and accurately indicated the role(s) that these authors had in your study. You can update author roles in the Author Contributions section of the online submission form.

2.2. Please also provide an updated Competing Interests Statement declaring this commercial affiliation along with any other relevant declarations relating to employment, consultancy, patents, products in development, or marketed products, etc.  

Reviewers' comments:

Reviewer's Responses to Questions

**Comments to the Author**

1. Is the manuscript technically sound, and do the data support the conclusions?

Reviewer #1: Partly

Reviewer #2: Partly

2. Has the statistical analysis been performed appropriately and rigorously? 

Reviewer #1: No

Reviewer #2: Yes

3. Have the authors made all data underlying the findings in their manuscript fully available?

Reviewer #1: Yes

Reviewer #2: Yes

4. Is the manuscript presented in an intelligible fashion and written in standard English?

Reviewer #1: No

Reviewer #2: Yes

5. Review Comments to the Author

Reviewer #1: Comments are mentioned in the attached pdf file. Although a nice work, statistics needs to be reviewed. A little bit of language errors. Histological images needs to be marked with arrows pointing to the damage or change seen.

Reviewer #2: The paper mentioned the use of polysorbate-20 as a replacement of the HSA in the BoNT/A formulation.

Major and minor correction required for the paper:

1. The study mentioned the use of Polysorbate-20 as an excipient for the BoNT products. But no where in the paper any results mention about presence of Botulinum neurotoxin along with polysorbate-20. According to study results polysorbate-20 is nontoxic excipient, and similar as HSA for the any pharmaceutical products administer through transdermal or intramuscular route. Not just limited for the BoNT. Additionally, Polysorbate-20 is already approved exicipient from FDA, so significance of this study is limited.

2. The title as well the introduction should be rewritten without emphasizing the Botulinum toxin. The current data seems the controls used for the formulation study. However, authors can include the possible use of polysorbated-20 as an excipient for Botulinum toxin products in the discussion but should not be only therapeutic product.

3. To use the polysorbated-20 as an excipient in the BoNT based formulation required in depth efficacy, safety, pharmacokinetics and pharmacodynamic study of the Botulinum toxin in the presence of polysorbate-20. The data should be compared with the existing formulation like Botox or dysport etc. The study having BoNT formulation with Polysorbate-20 has already published and mentioned in the paper as a reference # 18 and 19. So the significance of this particular control study reduces.

4. Data table 2 and 3 title are same. Please specify if table 2 belong to polysorbate-20 or for HSA.

6. PLOS authors have the option to publish the peer review history of their article (what does this mean?). If published, this will include your full peer review and any attached files.

Reviewer #1: No

Reviewer #2: No

---

## [Author Response · Author response to Decision Letter 0]

3 Jun 2021

[We have upload a separate file labeled 'Response to Reviewers'.]

Dear editor and reviewers,

We would like to thank the editor and the reviewers for helpful and constructive comments and critiques. We have done our best to revise the manuscript in order to satisfactorily address each of the points raised. We believe that this revised manuscript has greatly benefited from these changes and hope that it will be considered satisfactory for publication in PLOS ONE.

Response to Editor’s comment

- We are grateful to the editor for the comments and suggestions, which have helped us to improve our manuscript. We have revised the manuscript based on the editor’s comments and have provided our point-by-point responses to each comment below.

1. Why did you measure IgE levels and not for example HSA/PS specific IgG?

- Thank you for your comments. Generally, allergic reactions are divided into four types (Type I, II, III, and IV), of which type I and IV are the most common types of drug-related allergy (#1). Therefore, we conducted in vivo test, a standard method, to determine whether polysorbate 20 induces type I or IV allergies. Then, we confirmed that there was no allergic reaction, including abnormalities at the site of administration, due to polysorbate 20 in both studies (four-week repeated intramuscular dose toxicity study in SD rats and intradermal irritation study in NZW rabbits). After in vivo test, since only IgE-mediated anaphylaxis cases (type I allergy) were reported due to polysorbates (#2), we further measured the levels of serum IgE and histamine, which were type I allergy-related factors. The level of IgE and histamine also did not increase after administration of polysorbate 20 in both studies.

-#1: Böhm R, Proksch E, Schwarz T, Cascorbi I. Drug hypersensitivity: diagnosis, genetics, and prevention. Dtsch Arztebl Int. 2018;115: 501.

-#2: Maggio E. Polysorbates, biotherapeutics, and anaphylaxis: a review. Bioprocess Int. 2017.

2. What was the rationale for using the gastrocnemius muscle for injection rather than the thigh muscles?

- We thank you for pointing this out. In the case of BOTOX®, the first FDA-approved Botulinum toxin serotype A (BoNT/A) product, the route of administration was set to gastrocnemius muscle rather than thigh muscle in the toxicity study in rodents (Biologic license application number: 103000). In addition, in the compound muscle action potential (CMAP) and digit abduction scoring (DAS) techniques, which are generally used to evaluate the effectiveness of BoNT/A products, the route of administration was also set to gastrocnemius muscle rather than thigh muscle in the toxicity study in rodents (#1, 2, 3). Therefore, in the current study, the administration route was set to the gastrocnemius muscle for accurate comparison with the previously reported results. We have added the rationale for setting the route of administration to the revised manuscript.

-#1: Kim S-H, Kim S-B, Yang G-H, Rhee C-H. Mouse compound muscle action potential assay: An alternative method to conduct the LD50 botulinum toxin type A potency test. Toxicon. 2012;60: 341–347.

-#2: Aoki KR. A comparison of the safety margins of botulinum neurotoxin serotypes A, B, and F in mice. Toxicon. 2001;39: 1815–1820.

-#3: Oh H-M, Park JH, Song DH, Chung ME. Efficacy and safety of a new botulinum toxin type A free of complexing proteins. Toxins (Basel). 2016;8: 4.

3. We recommend rephrasing Lines 202-203 to removed the toxicologists' bias of looking for death in a study ("all animals survived..") to "..all animals remained well throughout the study and showed no clinical signs of disease, pain or distress.". Please also replace uses of the word 'symptoms' (what a patient reports) instead of 'signs' (what is observable).

- Thank you for your recommendation. As you suggest, we have revised the phrase “all SD rats survived throughout the study period” in the revised manuscript. In addition, we have modified the word ‘symptoms’ to ‘signs’ throughout the revised manuscript.

4. We note that the manuscript focuses mainly on the implications of this study for the development of cosmetics, and therefore request that you explicitly assert and emphasize the medical implications of this study in order to show the full range of applications of your findings.

- Thank you for your recommendation. As you suggest, we have modified the “Introduction” and “Conclusion” part accordingly in the revised manuscript. We have emphasized the significance of this study and mentioned the scope of application of our findings in the future.

Response to Reviewers' comments (Reviewer #1)

Comments are mentioned in the attached pdf file. Although a nice work, statistics needs to be reviewed. A little bit of language errors. Histological images needs to be marked with arrows pointing to the damage or change seen.

- Thank you for your encouraging feedback. Based on the comments in PDF file, we have revised the manuscript, including language errors and histological images. We have provided our point-by-point responses to each comment below.

1. Line 18: Modify "were" to "have". In addition, only one author can be the corresponding author.

- Thank you for pointing this out. We have modified the text accordingly in the revised manuscript.

- According to the PLOS ONE's submission guidelines, the following is stated in relation to the corresponding author: “Only one corresponding author can be designated in the submission system, but this does not restrict the number of corresponding authors that may be listed on the article in the event of publication. Whoever is designated as a corresponding author on the title page of the manuscript file will be listed as such upon publication. Include an email address for each corresponding author listed on the title page of the manuscript." Therefore, we kept two corresponding authors in the current study.

2. Line 25-27, 51-52, 80-81: Modify "few" to "a few".

-Thank you for pointing this out. We have modified the text accordingly in the revised manuscript.

3. Line 90 “Animals and Facility”: Mention the grouping of animals in this section. For example, Animals were divided into 5 groups with 11 animals in each group. Group 1 were administered 0.01 ml of substance x for y days and so on. Kindly do this for both the substances.

- Thank you for your comments. As you suggest, we have revised the text accordingly in the revised manuscript.

4. Line 95, 101: Provide weight range of animals

- Thank you for your comments. As you suggest, we have revised the text accordingly in the revised manuscript.

5. Line 101: Females are considered to be more sensitive. It would have been better if female animals were included. It will help in determining if there are any sex related differences.

- Thank you for pointing this out. In the ISO guideline, the following is stated in relation to the intradermal irritation study (#1): “Healthy young adult albino rabbits of either sex from a single strain, weighing not less than 2 kg, shall be used. A minimum of three animals shall initially be used to evaluate the test material. Inject intracutaneously 0,2 ml of the extract obtained with polar or non-polar solvent at five sites on one side of each rabbit.” Since the test substance was administered to five sites on the back skin, we selected a male as a test system, which was expected to have less interference between the administered substances because it is larger than that of female. In the current study, since there was no reaction by the administration of polysorbate 20 in the male SD rats, it is expected that there will be no side effects by the administration of polysorbate 20 in female SD rats.

-#1: Iso B, STANDARD B. Biological evaluation of medical devices. Part. 2009;1: 10993.

6. Were the rats and rabbits placed in the same animal room or different ones? Since the temperatures are different, is it that the mentioned temperatures are required for their survival?

- Thank you for your comments. The four-week repeated intramuscular dose toxicity study in SD rats were conducted in Medytox R&D Center (Republic of Korea) and all SD rats were purchased from Orient Bio, Inc. (Republic of Korea). The intradermal irritation study in NZW rabbits were performed at KTR (Republic of Korea), and all NZW rabbits were purchased from DooYeol Biotech (Republic of Korea). SD rats and NZW rabbits used in these two studies were placed in different facilities. In addition, the overall environment of the facilities for SD rats and NZW rabbits was optimally set based on the guidelines (#1 and #2)

-#1: ARRP. Guideline 20: Guidelines for the Housing of Rats in Scientific Institutions. Animal Research Review Panel. 2007.

-#2: ARRP. Guideline 18: Guidelines for the Housing of Rabbits in Scientific Institutions. Animal Research Review Panel. 2003

7. Line 178-179: Modify “based on” to “according to”.

- Thank you for pointing this out. We have modified the text accordingly in the revised manuscript.

8. Line 182-183: Delete “using the naked eye”.

- Thank you for pointing this out. We have modified the text accordingly in the revised manuscript.

9. Line 187-188: Modify “Then, all slides were” to “All slides were then” and “hematoxylin & eosin” to “hematoxylin and eosin”. In addition, delete “at KTR”. Avoid mentioning everytime that it was done at KTR. One time mention will suffice.

- Thank you for pointing this out. We have modified the text accordingly in the revised manuscript.

10. Line 227-230 “In SD rats, all substances were injected once every two weeks for four weeks (total three times; days 1, 15, 29) into one site of the gastrocnemius muscle of the right hindlimb.”: What was the basis for selection of these days for dosing. Why not daily as in acute toxicity studies?

- Thank you for your comments. The objective of this study was to verify the in vivo safety properties of polysorbate 20, an excipient of CORETOX® (newly developed BoNT/A product), when administered intramuscularly. Therefore, the dosing schedule (interval and frequency) was set based on the clinical application schedule of BoNT/A products (#1). The clinical application interval of BOTOX® (first FDA-approved BoNT/A product), which is similar to that of CORETOX®, is once every 3 months (Biologic license application number: 103000). In addition, according to the FDA approval document of BOTOX®, a 4-week repeated dose intramuscular toxicity study in rats was performed. Therefore, we have selected the dosing schedule of polysorbate 20 (once every two weeks for four weeks, total three times; days 1, 15, 29) based on clinical application dosing schedule, information of FDA-approved BoNT/A product, and ICH guideline. Furthermore, we thought that this four-week repeated intramuscular dose toxicity study could sufficiently replace the acute toxicity study. Because the dose of polysorbate 20 in the current study (0.02, 0.1, and 0.4 mg/kg) is more than six times the clinical application dose (0.003 ~ 0.06 mg/kg) (#2 and #3) and polysorbate 20 (with CORETOX®) is administered every 3 months (or single dose) in human, as mentioned above. Since the commented part is the figure legend section, we have modified the text accordingly in the method section.

-#1: Guideline ICHHT. Guidance on nonclinical safety studies for the conduct of human clinical trials and marketing authorization for pharmaceuticals M3 (R2). International conference on harmonisation of technical requirements for registration of pharmaceuticals for human use. 2009.

-#2: Lee J, Chun MH, Ko YJ, Lee S-U, Kim DY, Paik N-J, et al. Efficacy and safety of MT10107 (Coretox®) in post-stroke upper limb spasticity treatment: A randomized, double-blind, active drug-controlled, multi-center, phase III clinical trial. Arch Phys Med Rehabil. 2020.

-#3: Kwak S, Kang W, Rhee C-H, Yang G-H, Cruz DJM. Comparative Pharmacodynamics Study of 3 Different Botulinum Toxin Type A Preparations in Mice. Dermatologic Surg. 2020;46: e132–e138.

11. Line 231-232 “In NZW rabbits, all substances were injected once into five sites of the back skin (total one time; day 1).”: What is the basis of dosing selection?

- Thank you for your comments. As mentioned above (Q5), the following is stated in relation to the intradermal irritation study in the ISO guideline (#1): “Inject intracutaneously (intradermally) 0,2 ml of the extract obtained with polar or non-polar solvent at five sites on one side of each rabbit.” In addition, polysorbate 20 (with CORETOX®) is administered every 3 months (or single dose) in human, as mentioned above (Q10). Therefore, we have selected the dosing method of polysorbate 20 based on the ISO guideline and clinical application dosing schedule. Since the commented part is the figure legend section, we have modified the text accordingly in the method section.

12. Line 235-236, 332, Table 2 : Modify “hematology, clinical chemistry, serum IgE, and serum histamine” to “hematology, clinical chemistry and serum IgE and histamine”. Also, instead of chemistry, I feel the appropriate word used could be clinical biochemistry.

- Thank you for pointing this out. We have modified the text accordingly in the revised manuscript.

13. Line 239-240: Doses have already been mentioned in methods section. Repetition unnecessary. If any changes were observed, then mention the doses on which changes were observed only.

- Thank you for pointing this out. We have modified the text accordingly in the revised manuscript.

14. Line 241, 244: Significanlty higher or lower, please mention. In addition, modify “group administered” to “group that were administered”.

- Thank you for pointing this out. We have modified the text accordingly in the revised manuscript.

15. Line 242, 245, 284, 294, 342, 363: Delete “Vehicle”. Mention in the beginning of the manuscript that control group was administered the vehicle only. And then just mention control in all the places in the manuscript.

- Thank you for pointing this out. We have modified the text accordingly in the revised manuscript.

16. Line 248: If the doses are already mentioned in the methods section, there is no need to mention them again and again. It can be simply mentioned that no changes were seen in all the administered doses. If there is a change, them mention the specific dose at which the change was seen.

- Thank you for pointing this out. We have modified the text accordingly in the revised manuscript.

17. Line 253 and 256: Why arent the data represented as mean=/- standard error of mean? Furthermore, ANOVA or Kruskal-Wallis test are usually followed by a post hoc test such as Dunnet's test, Bonferoni's test or Tukey's test. However, none of the tests have been done to make comparasions. Authors kindly perform a test which best suits your data represented to make comparasions between test and control values and one test and another test values. Remove the shadings, significant values can be represented by a superscript *

- Thank you for pointing this out. The Kruskal-Wallis test performed to compare toxicological indicators in the current study is a ranking-based nonparametric statistical test. Therefore, we presented the median and range in Tables 2 and 3 to show the data without statistical bias. However, since the raw data of the information presented in Tables 2 and 3 are all presented in S3 table, we have modified Tables 2 and 3 (mean ± standard error of the mean, no shadings) accordingly in the revised manuscript. In addition, the Dunn's Rank Sum test was performed as a post hoc test of Kruskal-Wallis test for the significance of intergroup difference between the control and administered groups. We have modified the text accordingly in the method section.

18. Line 257: Modify "in" to "of".

- Thank you for pointing this out. We have modified the text accordingly in the revised manuscript.

19. Line 258: Modify "polysorbate 20 or HSA related macroscopic findings were not observed ~" to " no polyabsorbate 20 or....... macroscopic findings were observed such as changes in external appearance......".

- Thank you for pointing this out. We have modified the text accordingly in the revised manuscript.

20. Line 259-276: Changes observed should be represented as the mean or average changes observed in the whole group, not individual animal changes. Do not mention individual animal changes, rather describe the mean value obtained from the observed group. Don’t mention individual rat results, mention what was most commonly observed. Follow this for all the below results too. Please mention only the most common effects seen over all. Please refer some toxicity papers and observe how data are represented.

- Thank you your recommendations. In the current study, there was no significant histopathological change according to the administered substance in SD rats. Therefore, we have deleted individual changes and only represented the change of the whole groups for the concise histopathology results. We have modified the text accordingly in the revised manuscript.

21. Line 267, 364: Modify “administration group” to “administered group”.

- Thank you for pointing this out. We have modified the text accordingly in the revised manuscript.

22. Line 332-334: Mention p value. And Modify “a” to “the”.

- Thank you for pointing this out. We have modified the text accordingly in the revised manuscript.

23. Figure 2: Histological images needs to be marked with arrows pointing to the damage or change seen.

- Thank you for pointing this out. In Figure 2, macrophage infiltration and neovascularization were seen in all administered group (Fig 2a ~ l). In Figure 3, No histopathological findings were seen in all administered groups (Fig 3a ~ f). Therefore, we have modified the Figure 2 and Figure legend 2 accordingly in the revised manuscript.

Response to Reviewers' comments (Reviewer #2)

The paper mentioned the use of polysorbate-20 as a replacement of the HSA in the BoNT/A formulation.

- Thank you for your overall comments, which have substantially improved the manuscript. We have provided our point-by-point responses to each comment below.

1. The study mentioned the use of Polysorbate-20 as an excipient for the BoNT products. But no where in the paper any results mention about presence of Botulinum neurotoxin along with polysorbate-20. According to study results polysorbate-20 is nontoxic excipient, and similar as HSA for the any pharmaceutical products administer through transdermal or intramuscular route. Not just limited for the BoNT. Additionally, Polysorbate-20 is already approved exicipient from FDA, so significance of this study is limited.

- Thank you for your comments. Polysorbate 20 is an inactive ingredient for drug products that have already been approved by the FDA for auricular, intramuscular, intravenous, nasal, ophthalmic, oral, subcutaneous, topical, and vaginal administration (Drug Approvals and Databases). However, since it is mainly used as an excipient in tablet and ophthalmic solutions, most of the safety studies on polysorbate 20 have been conducted on oral toxicity and skin irritation (#1 and 2). There are few studies on the safety of polysorbate 20 when administered intramuscularly. Therefore, the current study was conducted to obtain in vivo safety data for intramuscular administration of polysorbate 20, which has little data currently accumulated. Findings of the current study suggest that polysorbate 20 could be used as a safe excipient for various pharmaceuticals administered intramuscularly. In addition, BoNT/A product (CORETOX®) that uses polysorbate 20 as an excipient have already been approved by the Ministry of Food and Drug Safety (Republic of Korea) and are on the market. In the approval process, in vivo safety and efficacy studies were already conducted in the presence of Botulinum neurotoxin along with polysorbate 20 and the relevant results were submitted to regulatory authorities. Therefore, in the current study, the safety of Botulinum neurotoxin along with polysorbate 20 was not investigated.

-#1: Moore J. Final report on the safety assessment of polysorbates 20, 21, 40, 60, 61, 65, 80, 81, and 85. J Am Coll Toxicol. 1984;3: 1–82.

-#2: Lanigan RS. Final report on the safety assessment of PEG-20 sorbitan cocoate; PEG-40 sorbitan diisostearate; PEG-2,-5, and-20 sorbitan isostearate; PEG-40 and-75 sorbitan lanolate; PEG-10,-40,-44,-75, and-80 sorbitan laurate; PEG-3, and-6 sorbitan oleate; PEG-80 sorb. Int J Toxicol. 2000;19: 43–89.

2. The title as well the introduction should be rewritten without emphasizing the Botulinum toxin. The current data seems the controls used for the formulation study. However, authors can include the possible use of polysorbated-20 as an excipient for Botulinum toxin products in the discussion but should not be only therapeutic product.

- Thank you for your recommendation. As you suggest, we have revised the title to “Safety verification for polysorbate 20, pharmaceutical excipient for intramuscular administration, in Sprague-Dawley rats and New Zealand White rabbits”. Throughout the ‘Introduction’ part, we have rewritten the manuscript to emphasize polysorbate 20 and HSA rather than Botulinum toxin. In addition, in ‘Introduction’ and ’Conclusions’ part, we have revised the manuscript to emphasize that the findings of the current study could be applied to a various therapeutic product, not limited to BoNT/A product.

3. To use the polysorbated-20 as an excipient in the BoNT based formulation required in depth efficacy, safety, pharmacokinetics and pharmacodynamic study of the Botulinum toxin in the presence of polysorbate-20. The data should be compared with the existing formulation like Botox or dysport etc. The study having BoNT formulation with Polysorbate-20 has already published and mentioned in the paper as a reference # 18 and 19. So the significance of this particular control study reduces.

- Thank you for your comments. As mentioned in the answer to Q2, in vivo safety, efficacy, pharmacodynamic studies were already conducted in the presence of Botulinum neurotoxin along with polysorbate 20 and the relevant results were submitted to regulatory authorities. Considering the characteristics of Botulinum toxin (local diffusion at the injection site, followed by rapid systemic metabolism and excretion), pharmacokinetic studies were not conducted. In those studies, the efficacy and toxicity of BOTOX® (uses HSA as an excipient) and CORETOX® (uses polysorbate 20 as an excipient) were also compared. In addition, since polysorbate 20 is an inactive ingredient, it is considered that efficacy, pharmacokinetics, and pharmacodynamic studies other than safety study are unnecessary when used alone. Therefore, the current study was conducted to obtain in vivo safety data of polysorbate 20. Although polysorbate 20 have already been approved by the FDA for the pharmaceutical excipient, the current study is considered worthwhile given the lack of safety data on intramuscular administration of polysorbate 20 (#1 and #2). It is expected that findings of the current study provides the basis for the in vivo safety profile of polysorbate 20 administered intramuscularly and the scientific reliability of the use of polysorbate 20 as an excipient for various pharmaceuticals in terms of its safety.

-#1: Moore J. Final report on the safety assessment of polysorbates 20, 21, 40, 60, 61, 65, 80, 81, and 85. J Am Coll Toxicol. 1984;3: 1–82.

-#2: Lanigan RS. Final report on the safety assessment of PEG-20 sorbitan cocoate; PEG-40 sorbitan diisostearate; PEG-2,-5, and-20 sorbitan isostearate; PEG-40 and-75 sorbitan lanolate; PEG-10,-40,-44,-75, and-80 sorbitan laurate; PEG-3, and-6 sorbitan oleate; PEG-80 sorb. Int J Toxicol. 2000;19: 43–89.

4. Data table 2 and 3 title are same. Please specify if table 2 belong to polysorbate-20 or for HSA.

- Thank you for your comments. Table 2 shows the level of hematology, clinical biochemistry, and serum IgE and histamine in male SD rats, while Table 3 shows the same values as table 2 in female SD rats.

---

## [Decision Letter · Decision Letter 1]

9 Aug 2021

PONE-D-21-03019R1

Safety verification for polysorbate 20, pharmaceutical excipient for intramuscular administration, in Sprague-Dawley rats and New Zealand White rabbits

PLOS ONE

Dear Dr. Kang,

Thank you for submitting your manuscript to PLOS ONE. After careful consideration, we feel that it has merit but does not fully meet PLOS ONE’s publication criteria as it currently stands. Therefore, we invite you to submit a revised version of the manuscript that addresses the points raised during the review process.

We look forward to receiving your revised manuscript.

Kind regards,

Yasmina Abd‐Elhakim

Academic Editor

PLOS ONE

Journal Requirements:

Reviewers' comments:

Reviewer's Responses to Questions

**Comments to the Author**

1. If the authors have adequately addressed your comments raised in a previous round of review and you feel that this manuscript is now acceptable for publication, you may indicate that here to bypass the “Comments to the Author” section, enter your conflict of interest statement in the “Confidential to Editor” section, and submit your "Accept" recommendation.

Reviewer #1: All comments have been addressed

Reviewer #2: All comments have been addressed

2. Is the manuscript technically sound, and do the data support the conclusions?

Reviewer #1: Yes

Reviewer #2: Yes

3. Has the statistical analysis been performed appropriately and rigorously? 

Reviewer #1: Yes

Reviewer #2: Yes

4. Have the authors made all data underlying the findings in their manuscript fully available?

Reviewer #1: Yes

Reviewer #2: Yes

5. Is the manuscript presented in an intelligible fashion and written in standard English?

Reviewer #1: No

Reviewer #2: Yes

**6. Review Comments to the Author**

**Reviewer #1: **All queries have been addressed. An interesting work. Some minor language errors. References require a little bit of checking. The comments are mentioned in the attached pdf file.

**Reviewer #2:** (No Response)

7. PLOS authors have the option to publish the peer review history of their article (what does this mean?). If published, this will include your full peer review and any attached files.

Reviewer #1: No

Reviewer #2: No

---

## [Author Response · Author response to Decision Letter 1]

16 Aug 2021

[We have upload a separate file labeled 'Response to Reviewers'.]

Dear editor and reviewers,

We appreciate the editor and the reviewers to review our manuscript. Our manuscript has been remarkably improved due to their valuable comments. Our responses to the comments are provided below.

Reviewers' comments

[Reviewer #1]

- Thank you for your feedback. We have provided our point-by-point responses to each comment below.

Q1. Abstract does not contain keywords?

- We have already submitted keywords within the submission system (Botulinum toxin type A products; Pharmaceutical excipient; Polysorbate 20; Human serum albumin). However, we have added keywords in the revised manuscript.

- Original version: N/A

- Revised version: [Line 52-53] Keywords: Botulinum toxin type A products, Pharmaceutical excipient, Polysorbate 20, Human serum albumin

Q2. Line 57: 19 superscript th

- Thank you for pointing this out. We have modified the text accordingly in the revised manuscript.

- Original version: [Line 57] Therefore, HSA has been used for therapeutic purposes in patients with hemorrhage, hypovolemia, and hypoalbuminemia since about the 19th century [2].

- Revised version: [Line 60] Therefore, HSA has been used for therapeutic purposes in patients with hemorrhage, hypovolemia, and hypoalbuminemia since about the 19th century [2].

Q3. Line 126-127 (Table 1): Number of animals (male/female)

- Thank you for pointing this out. We have modified the text accordingly in the revised manuscript.

- Original version: [Table 1] No. animals (Male/Female) 

- Revised version: [Table 1] Number of animals (male/female)

Q4. Line 131: Which

- Thank you for pointing this out. We have modified the text accordingly in the revised manuscript.

- Original version: [Line 131] HSA (Green Cross Corp., Republic of Korea) was selected as the comparative substance for the polysorbate 20 (Merck, Germany) that was the test substance of the current study.

- Revised version: [Line 135] HSA (Green Cross Corp., Republic of Korea) was selected as the comparative substance for the polysorbate 20 (Merck, Germany) which was the test substance of the current study.

Q5. Line 137-139: Delete ‘Then’ and ‘manner as when determining the dose of HSA’. Join this sentence (As a result, 0.02, 0.1, and 0.4 mg/kg were selected as the dose of polysorbate 20.) with the earlier sentence.

- Thank you for comments. We have modified the text accordingly in the revised manuscript.

- Original version: [Line 137-139] Then, the dose of polysorbate 20 was also calculated in the same manner as when determining the dose of HSA. As a result, 0.02, 0.1, and 0.4 mg/kg were selected as the dose of polysorbate 20.

- Revised version: [Line 141-142] The dose of polysorbate 20 was also calculated in the same manner and 0.02, 0.1, and 0.4 mg/kg were selected as the dose of polysorbate 20.

Q6. Line 189, 206: Delete ‘Then’. Please do not start a sentence with then. Kindly correct in other places also.

- Thank you for pointing this out. We have modified the text accordingly in the revised manuscript.

- Original version: [Line 41] Then, an intradermal irritation study was further conducted with 18 New Zealand White (NZW) rabbits.

[Line 111] Then, animals were divided into six groups with 11 animals (six male and five female SD rats) in each group. 

[Line 122] Then, animals were divided into six groups with 3 male NZW rabbits in each group.

[Line 160] Then, all substances were injected once into five sites (three left side and two right side) on the back skin (total 1 time; day 1) based on the guideline of the International Organization for Standardization (ISO) [24].

[Line 178] Then, the SD rats were anesthetized with isoflurane and blood samples were collected from their abdominal vein.

[Line 189] Then, the following parameters were examined in KTR using an automatic analyzer (TBA-120FR, TOSHIBA, Japan):

[Line 206] Then, the injection site (one site of the gastrocnemius muscle of the right hindlimb for the SD rats and the five sites of the back skin for the NZW rabbits) were collected and stored in 10% neutral buffered formalin.

- Revised version: [Line 41] An intradermal irritation study was further conducted with 18 New Zealand White (NZW) rabbits.

[Line 115] Animals were divided into six groups with 11 animals (six male and five female SD rats) in each group.

[Line 125] Animals were divided into six groups with 3 male NZW rabbits in each group.

[Line 165] All substances were then injected once into five sites (three left side and two right side) on the back skin (total 1 time; day 1) based on the guideline of the International Organization for Standardization (ISO) [24].

[Line 183] The SD rats were anesthetized with isoflurane and blood samples were collected from their abdominal vein.

[Line 194] The following parameters were then examined in KTR using an automatic analyzer (TBA-120FR, TOSHIBA, Japan):

[Line 211] The injection site (one site of the gastrocnemius muscle of the right hindlimb for the SD rats and the five sites of the back skin for the NZW rabbits) were collected and stored in 10% neutral buffered formalin.

Q7. Line 213: Please mention that data are represented as mean ± standard error of mean (SEM).

- Thank you for comments. We have modified the text accordingly in the revised manuscript.

- Original version: NA

- Revised version: [Line 227-228] Data are represented as mean ± standard error of the mean (SEM).

Q8. Line 232: Delete ‘Changes in’.

- Thank you for pointing this out. We have modified the text accordingly in the revised manuscript.

- Original version: [Line 232] Changes in body weights

- Revised version: [Line 238] Body weights

Q9. Line 235 and 264: In a sentence wherever the word table comes please make the ‘t’ small. While mentioning as a caption or a table then the ‘t’ can be capitalised. Kindly make these corrections in other locations as well. 

- Thank you for comments. According to the PLOS ONE's submission guidelines, the following is stated in relation to tables: 

- [Table citations] Tables should be cited as “Table 1”, “Table 2”, etc.

- [Examples] Vestibulum adipiscing urna ut lectus gravida, et bland Table 1.

- Therefore, we have kept the existing table citation form.

Q10. Line 237: Is it Fig. 1a or Fig 1a? Please kindly refer the journal guidelines once.

- Thank you for comments. According to the PLOS ONE's submission guidelines, the following is stated in relation to the figures:

- [Figure citations] Cite figures as “Fig 1”, “Fig 2”, etc.

- [Examples] Vestibulum adipiscing urna ut lectus gravida, vitae (Fig 1) interdum. Nam sit amet nulla lacus a, Figs 1 and 2 ultrices tellus.

- Therefore, we have kept the existing figure citation form.

Q11. Line 239, 241: “respectively” is inappropriate here, since you have already mentioned the weight immediately against the day. If days were mentioned consecutively followed by weights consecutively, then respectively may be used.

- Thank you for pointing this out. We have modified the text accordingly in the revised manuscript.

- Original version: [Line 237-241] Compared to the body weight before the first administration, all male SD rats had a body weight of 138.4% at day 8, 173.6% at day 15, 200.4% at day 22, and 225.0% at day 29, respectively, while all female SD rats had a body weight of 132.2% at day 8, 153.1% at day 15, 173.1% at day 22, and 187.3% at day 29, respectively.

- Revised version: [Line 243-246] Compared to the body weight before the first administration, all male SD rats had a body weight of 138.4% at day 8, 173.6% at day 15, 200.4% at day 22, and 225.0% at day 29, while all female SD rats had a body weight of 132.2% at day 8, 153.1% at day 15, 173.1% at day 22, and 187.3% at day 29.

Q12. Line 261-262: Delete “Findings of”. Delete “the level of” (hematology is a study not level). Change “serum IgE and histamine” to “serum IgE and histamine levels”.

- Thank you for pointing this out. We have modified the text accordingly in the revised manuscript.

- Original version: [Line 261-262] Findings of the level of hematology, clinical biochemistry, and serum IgE and histamine

- Revised version: [Line 267-268] Hematology, clinical biochemistry, and serum IgE and histamine levels

Q13. Line 266-272: Delete “was”. 

- Thank you for pointing this out. We have modified the text accordingly in the revised manuscript.

- Original version: [Line 266-272] In clinical biochemistry, compared to the control group, the albumin/globulin ratio was significantly decreased in male group 6 (p < 0.05), the level of glucose was significantly decreased in male group 3, 5, and 6 (p < 0.05, respectively), and the level of sodium was significantly increased in male group 4 (p < 0.01) and 6 (p < 0.05) (Table 2). 

In females, the level of total bilirubin was significantly decreased in group 2 and 4 (p < 0.05, respectively) and the level of chloride was significantly increased in group 2 (p < 0.05) and 3 (p < 0.01), compared to the control group (Table 3).

- Revised version: [Line 271-277] In clinical biochemistry, compared to the control group, the albumin/globulin ratio significantly decreased in male group 6 (p < 0.05), the level of glucose significantly decreased in male group 3, 5, and 6 (p < 0.05, respectively), and the level of sodium significantly increased in male group 4 (p < 0.01) and 6 (p < 0.05) (Table 2). 

In females, the level of total bilirubin significantly decreased in group 2 and 4 (p < 0.05, respectively) and the level of chloride significantly increased in group 2 (p < 0.05) and 3 (p < 0.01), compared to the control group (Table 3).

Q14. Line 277: Change “increase or decrease” to “change”.

- Thank you for pointing this out. We have modified the text accordingly in the revised manuscript.

- Original version: [Line 276-278] There was also no significant increase or decrease in the level of serum IgE and histamine in NZW rabbits administered polysorbate 20 and HSA compared to the control group.

- Revised version: [Line 282-284] There was also no significant change in the level of serum IgE and histamine in NZW rabbits administered polysorbate 20 and HSA compared to the control group.

Q15. Line 279 and 284: Delete “The level of”, and “in the current study”. Change “IgE and histamine” to “IgE and histamine levels”.

- Thank you for pointing this out. We have modified the text accordingly in the revised manuscript.

- Original version: [Line 279] Table 2. The level of hematology, clinical biochemistry, and serum IgE and histamine in male Sprague-Dawley (SD) rats in the current study.

[Line 284-285] Table 3. The level of hematology, clinical biochemistry, and serum IgE and histamine in female Sprague-Dawley (SD) rats in the current study.

- Revised version: [Line 285] Table 2. Hematology, clinical biochemistry, and serum IgE and histamine levels in male Sprague-Dawley (SD) rats.

[Line 290] Table 3. Hematology, clinical biochemistry, and serum IgE and histamine levels in female Sprague-Dawley (SD) rats.

Q16. Line 280-282 and 286-288: Change “Data were presented as mean ± standard error of the mean” to “Data ae presented as mean ± SEM”. Delete “All statistical analyses were performed using GraphPad Prism 282 8.0 (GraphPad Software, Inc., San Diego, CA)” 

- Thank you for pointing this out. We have modified the text accordingly in the revised manuscript.

- Original version: [Line 280-282] Data were presented as mean ± standard error of the mean. All data were analyzed using Kruskal-Wallis test and the significance of intergroup difference between the control and administered groups was assessed using Dunn's Rank Sum test. All statistical analyses were performed using GraphPad Prism 8.0 (GraphPad Software, Inc., San Diego, CA).

[Line 286-288] Data were presented as mean ± standard error of the mean. All data were analyzed using Kruskal-Wallis test and the significance of intergroup difference between the control and administered groups was assessed using Dunn's Rank Sum test. All statistical analyses were performed using GraphPad Prism 8.0 (GraphPad Software, Inc., San Diego, CA).

- Revised version: [Line 286-288] Data are presented as mean ± SEM. All data were analyzed using Kruskal-Wallis test and the significance of intergroup difference between the control and administered groups was assessed using Dunn's Rank Sum test. [deleted]

[Line 291-293] Data are presented as mean ± SEM. All data were analyzed using Kruskal-Wallis test and the significance of intergroup difference between the control and administered groups was assessed using Dunn's Rank Sum test. [deleted]

Q17. Line 290: Delete “Findings of”.

- Thank you for pointing this out. We have modified the text accordingly in the revised manuscript.

- Original version: [Line 290] Findings of necropsy and histopathology

- Revised version: [Line 295] Necropsy and histopathology

Q18. Line 371: Change “those” to “these”.

- Thank you for pointing this out. We have modified the text accordingly in the revised manuscript.

- Original version: [Line 371-372] Therefore, those findings were presumed to be due to the intramuscular injection rather than any effect of polysorbate 20.

- Revised version: [Line 376] Therefore, these findings were presumed to be due to the intramuscular injection rather than any effect of polysorbate 20.

Q19. Line 381: Just 2% will be better instead of 2.0%.

- Thank you for pointing this out. We have modified the text accordingly in the revised manuscript.

- Original version: [Line 379-382] However, despite its very low frequency, side effects in the allergy category have been reported in humans using shampoos (8.4% polysorbate 20, 2 cases among 5.88 million uses), cuticle softeners (2.0% polysorbate 20, 24 cases among 131 million use), and paste masks (2.0% polysorbate 20, 11 cases among 120.7 million uses) containing polysorbate 20 [14,15].

- Revised version: [Line 384-387] However, despite its very low frequency, side effects in the allergy category have been reported in humans using shampoos (8.4% polysorbate 20, 2 cases among 5.88 million uses), cuticle softeners (2% polysorbate 20, 24 cases among 131 million use), and paste masks (2% polysorbate 20, 11 cases among 120.7 million uses) containing polysorbate 20 [14,15].

Q20. Line 393-396: Change “hematology and clinical biochemistry” to “hematology and clinical biochemistry parameters”. Delete “of the NZW rabbits”.

- Thank you for pointing this out. We have modified the text accordingly in the revised manuscript.

- Original version: [Line 393-396] There were also no toxicologically significant changes, such as changes in the body weight or the hematology and clinical biochemistry in all rabbits administered polysorbate 20 and all toxicological values in NZW rabbits were within the normal range for the same age of the NZW rabbits [41,42].

- Revised version: [Line 398-401] There were also no toxicologically significant changes, such as changes in the body weight or the hematology and clinical biochemistry parameters in all rabbits administered polysorbate 20 and all toxicological values in NZW rabbits were within the normal range for the same age [41,42].

Q21. Line 397: Change “was” to “is”.

- Thank you for pointing this out. We have modified the text accordingly in the revised manuscript.

- Original version: [Line 396-398] These results of the intradermal irritation study indicated that polysorbate 20 was not an allergen in NZW rabbits, which was consistent with the results of the four-week repeated dose toxicity study in SD rats.

- Revised version: [Line 401-403] These results of the intradermal irritation study indicated that polysorbate 20 is not an allergen in NZW rabbits, which was consistent with the results of the four-week repeated dose toxicity study in SD rats.

Q22. Line 415: Authors maybe can wish to thank the institutions the works was carried out and the people who assisted in the work, if they wish to. Not applicable is not the appropriate term.

- Thank you for your comments. We have added thanks to colleagues who helped with the works.

- Original version: [Line 415] Not applicable

- Revised version: [Line 420-421] The authors thanks Do Yeon Lee and Min-Seo Choi, colleagues at the Medytox, Inc., for their helpful discussion and careful reading of the manuscript.

Q23. Line 466: Please check the journal abbreviation.

- Thank you for pointing this out. We have modified the text accordingly in the revised manuscript.

- Original version: [Line 464-466] Diehl K, Hull R, Morton D, Pfister R, Rabemampianina Y, Smith D, et al. A good practice guide to the administration of substances and removal of blood, including routes and volumes. J Appl Toxicol An Int J. 2001;21: 15–23.

- Revised version: [Line 470-472] Diehl K, Hull R, Morton D, Pfister R, Rabemampianina Y, Smith D, et al. A good practice guide to the administration of substances and removal of blood, including routes and volumes. J Appl Toxicol. 2001;21: 15–23.

Q24. Line 490 and 519: Please check the journal abbreviation (use standard abbreviations and check LTWA/ISSN website). 

- Thank you for your comments. According to the PLOS ONE's submission guidelines, the following is stated in relation to the reference format: 

- “PLOS uses the reference style outlined by the International Committee of Medical Journal Editors (ICMJE). Journal name abbreviations should be those found in the National Center for Biotechnology Information (NCBI) databases.” 

- We have checked journal name abbreviations (NCBI database) and kept the existing reference citation form.

[Reviewer #2]

- Thank you for the encouraging feedback, including previous reviews.

---

## [Editor Report · Decision Letter 2]

18 Aug 2021

Safety verification for polysorbate 20, pharmaceutical excipient for intramuscular administration, in Sprague-Dawley rats and New Zealand White rabbits

PONE-D-21-03019R2

Dear Dr. Kang,

We’re pleased to inform you that your manuscript has been judged scientifically suitable for publication and will be formally accepted for publication once it meets all outstanding technical requirements.

Kind regards,

Yasmina Abd‐Elhakim

Academic Editor

PLOS ONE
---

## [Editor Report · Acceptance letter]

20 Aug 2021

PONE-D-21-03019R2 

Safety verification for polysorbate 20, pharmaceutical excipient for intramuscular administration, in Sprague-Dawley rats and New Zealand White rabbits 

Dear Dr. Kang:

I'm pleased to inform you that your manuscript has been deemed suitable for publication in PLOS ONE. Congratulations! Your manuscript is now with our production department. 

Kind regards, 

on behalf of

Dr. Yasmina Abd‐Elhakim 

Academic Editor

PLOS ONE